# LEVERAGING FREE ENERGY IN PRETRAINING MODEL SELECTION FOR IMPROVED FINE-TUNING

## ABSTRACT

Recent advances in artificial intelligence have been fueled by the development of foundation models such as BERT, GPT, T5, and Vision Transformers. These models are first pretrained on vast and diverse datasets and then adapted to specific downstream tasks, often with significantly less data. However, the mechanisms behind the success of this ubiquitous pretrain-then-adapt paradigm remain underexplored, particularly the characteristics of pretraining checkpoints that lend themselves to good downstream adaptation. We introduce a Bayesian model selection criterion, called the downstream free energy, which quantifies a checkpoint's adaptability by measuring the concentration of nearby favorable parameters for the downstream task. We demonstrate that this free energy criterion can be effectively implemented without access to the downstream data or prior knowledge of the downstream task. Furthermore, we provide empirical evidence that the free energy criterion reliably correlates with improved fine-tuning performance, offering a principled approach to predicting model adaptability.

## 1 INTRODUCTION

The advent of foundation models has significantly reshaped the landscape of modern machine learning (Bommasani et al., 2021). Trained on expansive, diverse datasets using supervised or self-supervised learning methods, these models learn generalized representations that can then be successfully adapted (or finetuned) to a wide array of downstream tasks, often where there is significantly less data or limited computational resources (Bengio, 2012; Brown, 2020). This pretrain-then-adapt paradigm has emerged as a dominant and highly successful technique driving significant progress across natural language processing and computer vision with applications including text classification (Qiu et al., 2020), text generation (Li et al., 2024), image classification (Liu et al., 2023b), object detection (Sanchez et al., 2020), medical imaging (Mormont et al., 2018; Chen et al., 2019; Ke et al., 2021), autonomous driving (Kim & Park, 2017) and robotics (Jaquier et al., 2023).

As a result, there is a growing body of research aimed at better understanding the theoretical reasons behind the success of this pretrain-then-adapt paradigm (Galanti et al., 2022; Munn et al., 2024). One of the key open questions is to understand how to select pretraining checkpoints which are optimal for adaptation. A number of practical heuristics have emerged through experimental intuition and empirical analysis (Liu et al., 2023a), but a principled theoretical framework for effective checkpoint selection is still lacking.

To address this, we repurpose well-established concepts form Bayesian statistics and propose **downstream free energy** as a pretraining model selection criterion. Downstream free energy measures the negative log of the concentration of well-performing network weights near a pretraining checkpoint when evaluated on downstream data. Intuitively, lower downstream free energy indicates a higher concentration of parameters in parameter space for which the model is more adaptable and capable of generalizing well on downstream tasks. In short, checkpoints with lower downstream free energy are better suited for adaptation and thus should be preferred during pretraining.

Although the use of downstream free energy as a pretraining model selection criterion has strong theoretical motivations, it comes with an unfortunate caveat: to compute it requires access to the downstream dataset which may not be available to the practitioner during pretraining. However, under certain distributional shift conditions between the pretraining and downstream data, it is possible to overcome this limitation. Namely, we introduce the **pretraining free energy**, which is computed

solely on the pretraining data, and show that minimizing it serves as a reliable proxy for minimizing the downstream free energy (see Proposition 5.3). Together, these insights provide a solid justification for using the pretraining free energy as a model selection criterion during pretraining. This strategy is particularly advantageous when pretraining is intended to be general purpose, as is the case with most foundation models.

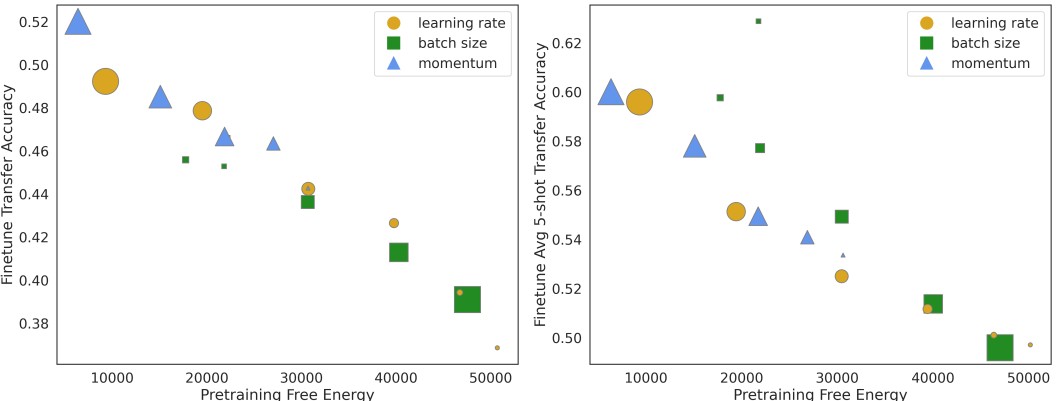

Figure 1: We plot pretraining free energy versus two types of transfer accuracy (left and right) for checkpoints at the end of pretraining. As expected, checkpoints with **lower pretraining free energy**, across various pretraining hyperparameters such as learning rate, batch size, and momentum, show **higher transfer accuracy**. The size of the icons represent magnitude of the hyperparameter value; e.g., a larger triangle means higher momentum. The reported values are averaged over five random seeds. For further details, refer to Section 6.

To justify our theoretical results, we exploit certain pretraining mechanisms that are known to reduce the pretraining free energy, such as larger learning rates, smaller batch sizes and higher momentum (Lau et al., 2023). We then verify that these mechanisms, which lead to reduced pretraining free energy, in turn correlate with improved downstream adaptation performance. A preview of these results is presented in Figure 1. To summarize, our contributions are as follows:

- We introduce the downstream free energy as novel model selection criterion for quantifying downstream adaptability; see Section 4.1.
- We show that the downstream free energy can be controlled by the pretraining free energy, under mild assumptions on the data distributions (Proposition 5.3), and provide insight into how this free energy perspective informs practical pretraining heuristics (Section 5.1).
- We verify experimentally that mechanisms which control the pretraining free energy indeed lead to improved downstream adaptability; see Section 6.

## 2 RELATIONSHIP TO PRIOR WORK

**Implicit bias in transfer learning.** The term implicit bias refers to the tendency of optimization processes, such as stochastic gradient descent (SGD), to inherently guide the model's learning dynamics towards solutions with properties which are not explicitly prescribed by the loss function (Neyshabur et al., 2017; Soudry et al., 2018; Gunasekar et al., 2018). For example, the selection of training hyperparameters, such as the learning rate and batch size, can have a significant effect on the optimization efficiency as well as on the quality of the learned model (Keskar et al., 2017; Masters & Luschi, 2018; Goyal, 2017; He et al., 2019; Andriushchenko et al., 2023). As a result, there has been considerable effort to understand the mechanisms which govern these implicit biases during model training. However, the effect of implicit bias in transfer learning—particularly how it impacts successful downstream domain adaptation—is a growing but less explored area of research (Lindsey & Lippl, 2023; Kumar et al., 2022).

In transfer learning, the ability to identify and leverage pretraining biases to predict and improve downstream test error is highly valuable. Recent work of Liu et al. (2023a); Galanti et al. (2022);

Munn et al. (2024) can be viewed as establishing relationships of the form

$$\text{downstream test error} \lesssim \text{pretraining characteristic.} \tag{1}$$

Ideally, these pretraining characteristics are sensitive to factors which can be manipulated by practitioners, thus allowing for deliberate influence and intentional design during pretraining. Furthermore, any such pretraining characteristic should be accessible using only pretraining data, since knowledge to the downstream task or data is typically not available. It is worthwhile to note that Liu et al. (2023a); Galanti et al. (2022); Munn et al. (2024) mainly consider the **linear probe** as their fine-tuning method while we consider full fine-tuning.

Liu et al. (2023a) explore the role of implicit bias in language modeling and establish an *empirical* relationship between the pretraining flatness (measured by the trace of the Hessian of the pretraining loss) and the downstream test accuracy. Their experiments verify that lower pretraining flatness, which they show is effectively regularized by SGD, strongly correlates with better downstream performance. Although this work does not provide a formal bound as in (1), it offers valuable empirical evidence on how the implicit flatness regularization of SGD acts to benefit transfer learning. This is particularly beneficial since techniques exist for explicitly minimizing loss landscape sharpness; e.g., Foret et al. (2020); Wen et al. (2023).

Galanti et al. (2022) examine the efficacy of transfer learning through the lens of neural collapse, a recently observed phenomenon which characterizes the geometry of last-layer features and weights for overparameterized classification networks (Papyan et al., 2020). They show through theory and experiments that the neural collapse exhibited during pretraining generalizes to new classes of the downstream task as well, thus enabling successful model adaptation. Drawing on the formalism described in (1), Galanti et al. (2022) can be seen as deriving theoretical bounds of the form

$$\text{downstream test error} \lesssim \text{downstream neural collapse} \lesssim \text{pretraining neural collapse.}$$

However, while this supports neural collapse as an effective pretraining characteristic, practical levers which can be used to explicitly regularize the pretraining neural collapse are lacking.

Munn et al. (2024) make progress in this direction by means of the geometric complexity, a model complexity measure introduced and analyzed in Dherin et al. (2022). They prove that the geometric complexity of the model's learned feature representations upper bounds the model neural collapse. Furthermore, their experiments verify that techniques which implicitly reduce this geometric complexity during pretraining (such as large learning rates, small batch sizes and increased $L^2$ regularization) in turn put regularizing pressure on the pretraining neural collapse leading to improved transfer test accuracy.

Our key contribution is the identification of *free energy* as a novel and significant pretraining characteristic which exhibits direct theoretical and empirical connections governing successful downstream model adaptability. We prove in Section 5 that, similar to neural collapse, the pretraining free energy bounds from above the downstream free energy. In addition, we establish (see Appendix A) a theoretical link between downstream free energy and the downstream Bayesian prediction, providing theoretical guarantees on the downstream Bayes test error. Together, these theoretical results, viewed in the context of (1), imply

$$\text{downstream Bayesian test error} \lesssim \text{downstream free energy} \lesssim \text{pretraining free energy.}$$

Furthermore, using mechanisms established in Lau et al. (2023) which are known to implicitly regularize the pretraining free energy—such as large learning rates, small batch sizes, and increased momentum—we experimentally verify (see Section 6) that lower pretraining free energy does indeed lead to improved fine-tuning performance.

**Bayesian model selection criterion.** The idea of using free energy has its roots in Bayesian model selection. Given a collection of models, $\mathcal{M}_1, \ldots, \mathcal{M}_k$, the task of choosing an optimal model for some given data is known as model selection. There are different (and sometimes irreconcilable) model selection criteria; but, in general, all model selection criteria attempt to balance fit and complexity. A particularly appealing Bayesian model selection criterion is the **free energy criterion** which is widely used and accepted in the both the statistical and machine learning literature (Hinton & van Camp, 1993; Kass & Raftery, 1995; MacKay, 2002; Robert et al., 2007). The free energy model selection criterion says we should pick the model with the lowest free energy. Since the free

energy is the negative log of the marginal likelihood, also known as Bayesian model evidence, free energy minimization is equivalent to marginal likelihood maximization. To our knowledge, this work represents the first application of the free energy criterion in the domain of transfer learning.

## 3 PROBLEM SETUP

We shall mainly treat the supervised setting though the theory developed below applies equally to the unsupervised setting. During pretraining, for input $x$ and target $y$, we employ a probabilistic model $p^0(y|x, w)$ parameterized by $w \in W \subset \mathbb{R}^p$. Throughout, we assume the pretraining model $p^0(y|x, w)$ depends on $x$ through a neural network $f_w^{\text{PT}}(x) = \sigma_{\text{out}}(v^T \phi_\theta(x))$ where $w = (v, \theta)$. Here $\phi_\theta$ denotes the feature extractor parameterized by $\theta$ and $v$ the weights of the linear head. The final activation is denoted $\sigma_{\text{out}}$; e.g., softmax or sigmoid for classification tasks.

For fine-tuning, we attach a new linear head $u$ to the backbone $\phi_\theta$ resulting in a neural network $f_{w'}^{\text{FT}}(x) = \sigma_{\text{out}}(u^T \phi_\theta(x))$ where $w' = (u, \theta)$ with $u$ potentially having different dimension to $v$. The fine-tuning probabilistic model is denoted $p^1(y|x, w')$ where the dependence on $x$ is through $f_{w'}^{\text{FT}}$.

Given a pretraining checkpoint $w^* = (v^*, \theta^*)$, we initialize $f_{w'}^{\text{FT}}$ at $(u_0, \theta^*)$ where $u_0$ is randomly initialized. All parameters of $w'$ are then fine-tuned via stochastic optimization. In this work, we employ **limited fine-tuning** where the linear head undergoes standard training, while the backbone remains mostly frozen, with updates governed by a separate, smaller learning rate. This approach is particularly useful in scenarios with limited downstream data, where the differential learning rates help to prevent overfitting or loss of general-purpose representations; cf. Lee et al. (2022).

For theoretical convenience, we will assume that $u$ and $v$ share the same dimensionality[1] This way, we can use $p(y|x, w)$ to denote both the pretraining and fine-tuning models. Let the true (and unknown) pretraining $(i = 0)$ and fine-tuning $(i = 1)$ joint distributions be denoted

$$r^i(x, y) := r^i(y|x)r^i(x), \quad i = 0, 1;$$

and define the pretraining $(i = 0)$ and fine-tuning $(i = 1)$ test loss to be

$$\mathrm{K}^i(w) := \mathbb{E}_{r^i(x)} D_{\text{KL}}(r^i(y|x) || p(y|x, w)).$$

Let $\mathcal{D}^0$ and $\mathcal{D}^1$ be datasets drawn from the pretraining and downstream distributions (resp.) and define the corresponding pretraining and fine-tuning sample losses to be

$$\hat{\mathrm{K}}^i(w) := \frac{1}{|\mathcal{D}^i|} \sum_{(x,y) \in \mathcal{D}^i} \left( \log r^i(y|x) - \log p(y|x, w) \right), \quad i = 0, 1.$$

Note that minimization of $\mathrm{K}^i(w)$ and $\hat{\mathrm{K}}^i(w)$ with respect to $w$ can recover the standard cross-entropy loss and squared loss frequently employed in deep learning. Indeed, if we drop the entropy term in $\mathrm{K}^i$ and $\hat{\mathrm{K}}^i$, which does not depend on $w$, we obtain the negative log likelihoods, for $i = 0, 1$,

$$\mathrm{L}^i(w) := -\mathbb{E}_{r^i(x,y)} \log p(y|x, w) \quad , \hat{\mathrm{L}}^i(w) := -\frac{1}{|\mathcal{D}^i|} \sum_{(x,y) \in \mathcal{D}^i} \log p(y|x, w).$$

We double load *test loss* to mean either $\mathrm{K}^i$ or $\mathrm{L}^i$ and *train loss* to mean either $\hat{\mathrm{K}}^i$ or $\hat{\mathrm{L}}^i$.

## 4 PRETRAINING AND DOWNSTREAM FREE ENERGY

In this section, we begin by introducing the **downstream free energy** as a measure of how suitable a checkpoint is for downstream adaptation. We then introduce the **pretraining free energy** as a proxy that can be measured solely using the pretraining data.

Let $U_0 = \{w_\alpha^* = (v_\alpha^*, \theta_\alpha^*)\}_\alpha$ denote the set of local minima of the pretraining test loss $\mathrm{K}^0(w)$. In our theoretical development, we will frequently refer to the elements of $U_0$ as *pretraining checkpoints*.

---

[1]This eases exposition by avoiding having to distinguish between $p^0(y|x, w)$ and $p^1(y|x, w')$. Note, we do not adhere to this restriction in our experiments.

The elements of $U_0$ are, however, distinct from the actual pretraining checkpoints that one obtains during the course of pretraining using the empirical loss $\hat{K}^0(w)$.

Given a *single* model – a parametric family $\mathcal{M} = \{p(y|x, w) : w \in W\}$ – with multiple optima (as neural networks are prone to exhibit), we can perform *internal model selection* (Balasubramanian, 1996) using a local version of the free energy criterion to select among the local optima. This amounts to comparing the downstream free energies between elements of $U_0$. We now give a precise definition of the downstream free energy associated to an element of $U_0$.

## 4.1 DOWNSTREAM FREE ENERGY

With datasets $\mathcal{D}^0$ and $\mathcal{D}^1$ as above, let $n = |\mathcal{D}^0|$ and $m = |\mathcal{D}^1|$. Informally, we might say that a pretraining checkpoint $w^* = (v^*, \theta^*) \in U_0$ is a good candidate for adaptation if there are many weights $\theta$ in the vicinity of $\theta^*$ with low fine-tuning test loss; i.e., low values of $K^1(w)$. One way to make this mathematically precise is through the **downstream free energy**

$$\bar{F}^1(B_\gamma(w^*)) := -\log \bar{Z}^1(B_\gamma(w^*)), \tag{1}$$

which is the negative log of a local marginal likelihood

$$\bar{Z}^1(B_\gamma(w^*)) := \int_{B_\gamma(w^*)} \exp\{-mK^1(w)\}\varphi(w)\, dw. \tag{2}$$

Here $\varphi(w)$ is a prior over the model parameters $w$, and $B_\gamma(w^*) := \{w = (v^*, \theta) : ||\theta - \theta^*||_2^2 \le 1/\gamma\}$ is the $\gamma$-neighborhood around $w^*$ with $v^*$ frozen. Note that large values of $\gamma$ force us to stay near $\theta^*$ and thus, ultimately, stay near the pretraining checkpoint $w^* = (v^*, \theta^*)$ as well.

Taken together, equations equation 1 and equation 2 imply that a large concentration of weights $\theta$ near $\theta^*$ with low downstream test loss $K^1(w)$ results in a large $\bar{Z}^1(B_\gamma(w^*))$ and, equivalently, a small $\bar{F}^1(B_\gamma(w^*))$. Thus, we propose the following **downstream free energy strategy** for improved fine-tuning:

> *Pretraining checkpoints with lower downstream free energy are more likely to adapt successfully to downstream tasks.*

Formally, we seek to find parameters $w^* \in U_0$ which minimize the downstream free energy; i.e.,

$$\arg\min_{w^* \in U_0} \bar{F}^1(B_\gamma(w^*)). \tag{3}$$

Before addressing the implementation of this free energy strategy, let's first understand the competing forces behind this model selection criterion. Given $w^* \in U_0$, following the techniques set out in Watanabe (2009), we can write the asymptotic expansion of $\bar{F}^1(B_\gamma(w^*))$ in the sample size $m$ as

$$\bar{F}^1(B_\gamma(w^*)) = mK^1(w^{*1}) + \lambda^1(w^*) \log m + O(\log\log m), \tag{4}$$

where

$$w^{*1} := \arg\min_{w \in B_\gamma(w^*)} K^1(w).$$

**Remark 4.1.** *From equation 4, note that that downstream free energy of a checkpoint $w^*$ is a weighted sum of two things: the fit, as measured by $K^1(w^{*1})$, and the complexity, as measured by $\lambda^1(w^*)$. This complexity measure $\lambda^1(w^*)$ was recently introduced as the **local learning coefficient**; see Lau et al. (2023). Lower local learning coefficient means lower model complexity. Note that a checkpoint with higher loss under the downstream distribution may still be preferred as long as its complexity is low enough to compensate. Furthermore, note that for pretraining checkpoints that are in the same level set of $K^1$, the checkpoint with the lowest model complexity, as measured by $\lambda^1$, will have the lowest downstream free energy.*

The free energy strategy in equation 3 which uses $\bar{F}^1(B_\gamma(w^*))$ to select among candidate checkpoints in $U_0$ is conceptually sound but presents two notable implementation challenges. First, $\bar{F}^1(B_\gamma(w^*))$, besides involving some unknown terms such as $K^1$, is the negative log of an intractable integral. This is not insurmountable as many techniques such as MCMC or variational inference are available to deal with intractable integrals.

The second, and more significant, issue is that applying $\bar{\mathrm{F}}^1(B_\gamma(w^*))$ to select among checkpoints $w^* \in U_0$ requires access to downstream data. This poses a problem because, in many practical scenarios, the downstream task may not be known or fully available during pretraining. To address this limitation, we introduce the pretraining free energy, an analog of the downstream free energy but which can be computed using only the pretraining data. In Section 5 we show how these two quantities are related.

### 4.2 PRETRAINING FREE ENERGY

Similar to the downstream free energy defined in equation 1, we define the **pretraining free energy** for a pretraining checkpoint $w^* = (v^*, \theta^*) \in U_0$ as

$$\mathrm{F}^0(B_\gamma(w^*); \beta) := -\log \mathrm{Z}^0(B_\gamma(w^*); \beta) \tag{5}$$

where

$$\mathrm{Z}^0(B_\gamma(w^*); \beta) := \int_{B_\gamma(w^*)} \exp\{-n\beta \hat{\mathrm{K}}^0(w)\} \varphi(w) \, dw \tag{6}$$

and $\beta > 0$ is an inverse temperature. Unlike $\bar{\mathrm{Z}}^1(B_\gamma(w^*))$ and $\bar{\mathrm{F}}^1(B_\gamma(w^*))$, here the quantities $\mathrm{Z}^0(B_\gamma(w^*); \beta)$ and $\mathrm{F}^0(B_\gamma(w^*); \beta)$ are stochastic. We indicate this by dropping the overhead bar.

Analogous to equation 4, the asymptotic expansion of $\mathrm{F}^0(B_\gamma(w^*); \beta)$ in $n$ for $w^* \in U_0$ is

$$\mathrm{F}^0(B_\gamma(w^*); \beta) = n\beta \hat{\mathrm{K}}^0(w^{*0}) + \lambda^0(w^*) \log n + O_p(\log \log n) \tag{7}$$

where

$$w^{*0} := \arg \min_{w \in B_\gamma(w^*)} \mathrm{K}^0(w).$$

Note that the asymptotic expansion of $\bar{\mathrm{F}}^1(B_\gamma(w^*))$ in equation 4 involves the downstream *test* loss $\mathrm{K}^1$ whereas the asymptotic expansion of $\mathrm{F}^0(B_\gamma(w^*); \beta)$ in equation 7 involves the pretraining *train* loss $\hat{\mathrm{K}}^0$. To compare the two, we take the expectation over the dataset in equation 7, arriving at the following expansion involving only deterministic quantities:

$$\mathbb{E}_{\mathcal{D}^0} \mathrm{F}^0(B_\gamma(w^*); \beta) = n\beta \mathrm{K}^0(w^{*0}) + \lambda^0(w^*) \log n + O(\log \log n). \tag{8}$$

In the next section, we will use these asymptotic expansions to bound the discrepancy between the downstream and pretraining free energy.

## 5 RELATIONSHIP BETWEEN PRETRAINING AND DOWNSTREAM FREE ENERGY

In this section, we show there is a satisfying relationship between pretraining free energy and downstream free energy, asymptotically speaking. Relying on the leading order terms of the asymptotic expansion of the downstream free energy in equation 4, we can express the downstream free energy strategy in equation 3 as

$$\arg \min_{w^* \in U_0} \left[ m\mathrm{K}^1(w^{*1}) + \lambda^1(w^*) \log m \right] \quad \text{where } w^{*1} := \arg \min_{w \in B_\gamma(w^*)} \mathrm{K}^1(w). \tag{9}$$

To avoid requiring the downstream test loss $\mathrm{K}^1$, we introduce the **pretraining asymptotic free energy strategy** which relies only on the pretraining distribution and (under mild assumptions, below) serves as a viable proxy for equation 9. Formally, this strategy seeks a solution of the following optimization

$$\arg \min_{w^* \in U_0} \left[ n\beta_0 \mathrm{K}^0(w^*) + \lambda^0(w^*) \log n \right] \quad \text{where } \beta_0 = M \frac{m \log n}{n \log m}. \tag{10}$$

Let us first state our underlying assumptions.

**Assumption 5.1.** *The parameter $\gamma$ is such that $w^{*0}$ is a local minimum of $\mathrm{K}^0(w)$; i.e., $w^{*0} \in U_0$.*

**Assumption 5.2.** *The pretraining distributions $r^0(x, y)$ and the downstream distribution $r^1(x, y)$ are such that*

$$M := \max_{(x,y) \sim r^0(x,y)} \frac{r^1(x, y)}{r^0(x, y)} < \infty.$$

Assumption 5.1 stipulates that the $\gamma$-neighborhood around $w^*$ is not too big so as to ensure that $w^*$ remains a local minimum in $B_\gamma(w^*)$. Assumption 5.2 stipulates that the pretraining and downstream distributions should not be too different; the same assumption was made in Yamazaki et al. (2007) for the purpose of study distribution shift.

**Proposition 5.3.** *Let $w^*$ be a local minimum of $\mathrm{K}^0(w)$; i.e., $w^* \in U_0$. Suppose Assumptions 5.1 and 5.2 hold. Further suppose $\lambda^1(w^*) \leq \lambda^0(w^*)$. Then we have*

$$\mathrm{K}^1(w^{*1}) + \lambda^1(w^*)\frac{\log m}{m} \leq M\mathrm{K}^0(w^*) + D + \lambda^0(w^*)\frac{\log m}{m} \tag{11}$$

*where $D = \int \log \frac{r^1(y|x)}{r^0(y|x)} r^1(x,y)\, dx\, dy$.*

*Proof.* By definition of the test loss and rearranging terms via change of measure, for all $w$,

$$\mathrm{K}^1(w) = \int \log\left(\frac{r^1(y|x)}{p(y|x,w)}\right) r^1(x,y) dx dy = \int \log\left(\frac{r^0(y|x)}{p(y|x,w)}\frac{r^1(y|x)}{r^0(y|x)}\right)\frac{r^1(x,y)}{r^0(x,y)} r^0(x,y) dx dy$$

$$= \int \log\left(\frac{r^0(y|x)}{p(y|x,w)}\right)\frac{r^1(x,y)}{r^0(x,y)} r^0(x,y)\, dx\, dy + \int \log\left(\frac{r^1(y|x)}{r^0(y|x)}\right) r^1(x,y)\, dx\, dy$$

$$\leq M\mathrm{K}^0(w) + D.$$

Also, by definition of $w^{*1}$, we have $\mathrm{K}^1(w^{*1}) \leq \mathrm{K}^1(w^*)$. Combining these two facts, we get $\mathrm{K}^1(w^*) \leq M\mathrm{K}^0(w^*) + D$ and obtain the conclusion in equation 11. $\qquad\square$

Proposition 5.3 justifies model selection using the asymptotic expansion of the pretraining free energy as in equation 10. This follows from equation 11 by first multiplying both sides by $m$ and then noting that minimizing $mM\mathrm{K}^0(w^*) + mD + \lambda^0(w^*)\log m$ is equivalent, up to constants, to minimizing $\frac{\log n}{\log m}\left[mM\mathrm{K}^0(w^*) + \lambda^0(w^*)\log m\right]$, which leads us precisely to equation 10. To further illustrate Proposition 5.3, we include explanatory examples in Appendix C which interprets this result applied to Gaussian distributions.

**Interpretation and Feasibility of Assumption 5.2** A natural question arises as to the feasibility of Assumption 5.2 and indeed this assumption may not always hold for some real-world scenarios. For example, if the pretraining data includes only images of horses while the downstream data contains only cars, their label supports would be disjoint, and thus violate Assumption 5.2. To address this, our experiments in Section 6 focus on settings where the pretraining dataset is significantly larger and more diverse than the downstream dataset. This also reflects common practice in the field and an established heuristic in transfer learning; see also (Kornblith et al., 2019). Specifically, we achieve this by using pretraining datasets with a substantially larger set of image classes. If this were reversed; i.e., the pretraining dataset has substantially fewer classes than the downstream dataset, the relationship we establish in Proposition 5.3 would be uninformative.

### 5.1 Observations of the pretraining asymptotic free energy strategy

In this section, we present practical observations that follow from selecting pretraining checkpoints according to the pretraining asymptotic free energy strategy defined by equation 10.

**Observation 1: A suboptimal checkpoint in terms of pretraining test loss can still be preferred by the pretraining asymptotic free energy strategy in equation 10.** Suppose we have two models $w_\alpha^*, w_\beta^* \in U_0$; i.e., both models are local minima of the pretraining test loss $\mathrm{K}^0$. In order to determine which model is preferred for fine-tuning, our strategy equation 10 directs us to compare $F_\alpha = n\beta_0\mathrm{K}^0(w_\alpha^*) + \lambda^0(w_\alpha^*)\log n$ and $F_\beta = n\beta_0\mathrm{K}^0(w_\beta^*) + \lambda^0(w_\beta^*)\log n$.

Suppose $\mathrm{K}^0(w_\alpha^*) < \mathrm{K}^0(w_\beta^*)$; i.e., $w_\alpha^*$ and $w_\beta^*$ are in different level sets and checkpoint $w_\alpha^*$ has lower pretraining test loss; but $\lambda^0(w_\alpha^*) > \lambda^0(w_\beta^*)$, implying checkpoint $w_\beta^*$ is less complex than checkpoint $w_\alpha^*$. Then it is entirely possible for $F_\alpha > F_\beta$ so that checkpoint $w_\beta^*$ will be preferred by equation 10 despite having higher pretraining test loss. In fact, this happens precisely when $\frac{m}{\log m} < \frac{1}{M}\frac{\lambda^0(w_\alpha^*) - \lambda^0(w_\beta^*)}{\mathrm{K}^0(w_\beta^*) - \mathrm{K}^0(w_\alpha^*)}$. Recall, $m$ represents the number of examples in the downstream dataset.

Note that, when $M$ is large, there's a smaller range of $m$ under which the suboptimal pretraining checkpoint will be preferred. In other words, if the downstream distribution is very different to the pretraining distribution, the free energy strategy will look to the lower level sets of pretraining test loss.

**Observation 2: When $n\beta_0 \gg \log n$, a checkpoint with lower pretraining test loss will always be preferred by the pretraining asymptotic free energy strategy in equation 10.** Again, suppose we have two local minima $w_\alpha^*, w_\beta^* \in U_0$ but which are in different level sets of the test loss; i.e., $K^0(w_\alpha^*) \neq K^0(w_\beta^*)$. Without any knowledge or bound on $\beta_0$, we cannot decide which checkpoint has lower free energy since, as described above in Observation 1, the complexity term $\lambda^0$ also plays a role in comparing $F_\alpha$ and $F_\beta$.

However, when $n\beta_0$ is significantly larger than $\log n$, the first term in equation 10 dominates the second. In this case, the pretraining asymptotic free energy strategy prioritizes checkpoints with lower pretraining test loss $K^0$.

Using the definition of $\beta_0$ in equation 10, the setting described here is equivalent to $Mm \gg \log m$, where $m$ is the size of the fine-tuning dataset and $M$ measures distributional shift. Since $m$ already grows faster than $\log m$, this may offer an intriguing insight which justifies the reliability of pretraining test loss as a heuristic for checkpoint adaptability.

**Observation 3: For checkpoints with the same pretraining test loss, the one with the lowest complexity is preferred by the pretraining asymptotic free energy strategy in equation 10.** Suppose we have two models $w_\alpha^*, w_\beta^* \in U_0$ in the same level set of $K^0$; i.e., they have the same pretraining test loss $K^0(w_\alpha^*) = K^0(w_\beta^*)$. As before, our strategy equation 10 directs us to compare $F_\alpha = n\beta_0 K^0(w_\alpha^*) + \lambda^0(w_\alpha^*)\log n$ and $F_\beta = n\beta_0 K^0(w_\beta^*) + \lambda^0(w_\beta^*)\log n$. However, since the first terms are equal, selecting the preferred pretraining checkpoint depends only on the model complexity, as measured by $\lambda^0(w_\alpha^*)$ and $\lambda^0(w_\beta^*)$. Therefore, holding all else the same, the strategy in equation 10 naturally prefers simple pretraining checkpoints over more complex ones for improved fine-tuning.

## 5.2 ESTIMATING PRETRAINING FREE ENERGY

So far, we have established the pretraining asymptotic free energy strategy as a theoretically principled approach to pretraining model selection for improved finetuning. In this section, we show how to estimate the pretraining asymptotic free energy required in equation 10 using only the sample pretraining train loss $\hat{L}^0$. This estimation technique, which we employ in our experiments (Section 6), enables the application of our proposed strategy in equation 10 for real-world machine learning scenarios.

We begin by focusing first on model selection for pretraining checkpoints in the same level set of $K^0$. In this case, we can set $\beta_0$ to an arbitrary value; we set $\beta_0 = 1$. Next, note that the optimization objective in equation 10 can be equivalently expressed in terms of $L^0$ since it differs only from $K^0$ by a constant with respect to $w$. In other words, we have

$$\arg \min_{w^* \in U_0} \left[ nK^0(w^*) + \lambda^0(w^*)\log n \right] = \arg \min_{w^* \in U_0} \left[ nL^0(w^*) + \lambda^0(w^*)\log n \right]. \quad (12)$$

To estimate the RHS of equation 12, we refer to recent work of Lau et al. (2023) which shows that the Widely Applicable Bayesian Information Criterion (WBIC) around $w^* \in U_0$ is an asymptotically unbiased estimator of $nL^0(w^*) + \lambda^0(w^*)\log n$. This localized version of the WBIC is computed from the sample pretraining train loss $\hat{L}^0$ measured in the neighborhood $B_\gamma(w^*)$ of the checkpoint $w^*$ as described below.

Consider a localizing Gaussian prior which acts as a surrogate for enforcing the domain of integration given by $B_\gamma(w^*)$. Specifically, let $\varphi_{\vec{\gamma}}(w) \propto \exp\{-\vec{\gamma}^T ||w||_2^2\}$, $\quad \vec{\gamma} \in \mathbb{R}_{>0}^p$ which is centered at the origin with scale vector $\vec{\gamma} = (\gamma_1, \ldots, \gamma_p)$. Since we only want to measure the free energy with respect to parameters $\theta$ of the model backbone (recall, the fine-tuning setup described in Section 3), we take $\gamma_j = \infty$ in the coordinates of $v$ and $\gamma_j = \gamma$ in the coordinates of $\theta$, where $\gamma$ is the same as the radius defining the neighborhood $B_\gamma(w^*)$; recall, equation equation 2.

Define the pretraining posterior distribution

$$p^0(w; w^*, \beta, \vec{\gamma}) \propto \exp\{-n\beta\hat{L}^0(w)\}\varphi_{\vec{\gamma}}(w - w^*). \tag{13}$$

Following Lau et al. (2023), we define the **pretraining WBIC** at $w^* \in U_0$ by

$$\text{WBIC}(w^*; \beta^*) := \int \left[ n\hat{L}^0(w) \right] p^0(w; w^*, \beta^*, \gamma) \, dw, \tag{14}$$

where $\beta^* = \frac{1}{\log n}$. The pretraining WBIC at a checkpoint $w^*$ is a good estimate of the (expected) pretraining free energy around $w^*$ defined by equations equation 5 and equation 6. Furthermore, $\text{WBIC}(w^*; \beta^*)$ can be reliably computed using SGLD sampling methods; see (Lau et al., 2023, Appendix G: Algorithm 1).

Therefore, to apply the pretraining asymptotic free energy strategy in equation 10 to check-points with the same $K^0$, we simply select the one with the smallest pretraining WBIC given by $\text{WBIC}(w^*; \beta^*)$. In the next section, we empirically verify this strategy using the CIFAR dataset trained on ResNet-18.

## 6 EXPERIMENTS

The goal of our experiments is to evaluate how well the pretraining WBIC, which estimates the pretraining free energy as described in Section 5.2, correlates with downstream performance. In order to measure the impact of lower pretraining WBIC, we apply mechanisms during pretraining which are known to implicitly regularize this quantity, as shown in (Lau et al., 2023). These include including large learning rates, small batch sizes, and high momentum.

We use the CIFAR-FS dataset (Bertinetto et al., 2019), derived from CIFAR-100 where the 100 classes are divided into 64 classes for meta-training, 16 classes for meta-validation, and 20 classes for meta-testing. We pretrain on the meta-training set and then assess model adaptability on the unseen meta-test set via limited fine-tuning described in Section 3. The meta-validation classes are not used.

**Pretraining.** For pretraining, we use all 64 classes from the CIFAR-FS meta-training set to train a ResNet-18 model using stochastic gradient descent (SGD). We explore ranges of hyperparameter values for the learning rate, batch size and momentum. Interaction effects between these are not considered. Full experiment details for each hyperparameter sweep are provided in Appendix D.1. During training we track the pretraining train loss (first column of Figure 2) and the pretraining WBIC (second column of Figure 2). The hyperparameter settings for pretraining WBIC computation are provided in Appendix D.1.

**Full meta-test fine-tuning** uses the full meta-test dataset, consisting of all 20 meta-test classes with 600 examples per class. We use an 80/20 split for training and testing, with stratification within each class. In this setting a new (randomly initialized) linear head is attached for the 20-class classification task, and the model is fine-tuned for 100 steps using SGD. This setting corresponds to the "Finetune Transfer Accuracy" metric (third column) in Figure 2. Hyperparameter details for this setting are in Appendix D.2.

**Few-shot meta-test fine-tuning** examines a data-limited, few-shot scenario. A single few-shot task is created by randomly sampling 5 classes and 5 examples per class from the meta-test dataset, creating a dataset with 25 total training examples. A new (randomly initialized) linear head is attached for the 5-class classification task, and the model is finetuned for 100 steps using full batch gradient descent. The transfer accuracy is evaluated on 100 randomly selected test examples for each of the 5 classes. The overall transfer accuracy is averaged over 100 few-shot tasks. This setting corresponds to the "Avg 5-shot Transfer Accuracy" metric (fourth column) in Figure 2. Hyperparameter details for this setting are in Appendix D.2.

**Results.** In each of these two fine-tuning scenarios, we observe a strong correlation between lower pretraining free energy (as measured by the pretraining WBIC, see Section 5.2) and better down-stream performance; see Figure 2. In particular, we see that increasing learning rate, decreasing batch sizes, and increasing momentum all result in lower pretraining WBIC, which in turn leads to better downstream performance. Note the Avg 5-shot transfer accuracy (fourth column) is typically higher than the finetune transfer accuracy (third column); this is likely because the former

only needs to learn 5 classes at a time while the latter needs to learn 20 classes. Interestingly, we can view pretraining train loss (the first column of Figure 2) as a baseline comparison. We see that pretraining train loss often collapses to a similar value as training proceeds, rendering it ineffective for distinguishing different fine-tuning behaviors.

In Figure 1, we take each checkpoint at the end of pretraining and plot its pretraining WBIC (called pretraining free energy there since the terminology had not been introduced) versus transfer accuracy. The left (right) plot of Figure 1 corresponds to the third (fourth) column of Figure 2.

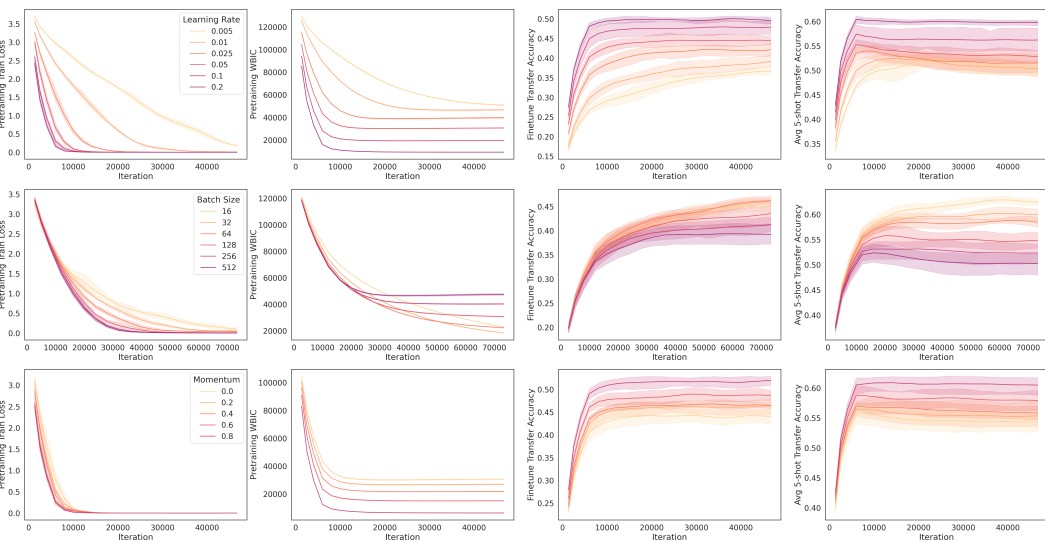

Figure 2: Model checkpoints with lower pretraining WBIC (second column) consistently result in better transfer accuracy, both when fine-tuning on the full downstream dataset (third column) and in the few-shot setting (fourth column). Lower pretraining WBIC correlates with better downstream performance for **Top row:** larger learning rates, **Middle row:** smaller batch sizes, and **Bottom row:** increased momentum.

# 7 CONCLUSION AND LIMITATIONS

In this work, we introduced the downstream free energy as a Bayesian model selection criterion for quantifying the adaptability of pretraining checkpoints, offering a principled way to predict their performance on unseen downstream tasks. Our key insight is that checkpoints with lower downstream free energy are more adaptable, making them ideal candidates for fine-tuning. Our empirical results validate the utility of the pretraining free energy as a practical checkpoint selection criterion, especially in scenarios where downstream data is scarce or inaccessible.

Despite the promising results, some limitations remain. First, our analysis currently lacks a direct link between downstream free energy and downstream predictive performance. At the moment, we provide a rigorous connection only when downstream adaptation is performed in a *Bayesian* manner (see Appendix A). However, while Bayesian deep learning is not yet widely adopted due to its computational overhead, this link may become valuable as computational barriers are reduced, particularly in fine-tuning scenarios.

In addition, while our theoretical framework supports the use of free energy as a selection criterion, the practical computation of free energy, in the form of the pretraining WBIC as in equation 14, remains challenging for large models. An alternative approach may involve identifying computationally efficient "levers" that influence pretraining free energy, allowing us to improve downstream adaptation performance without relying on direct computation of the pretraining WBIC.

## 8 REPRODUCIBILITY STATEMENT

To facilitate the reproduction of our results, we have provided comprehensive details of our experimental setup, including training procedures, data processing steps, and hyperparameter settings (e.g., learning rate, batch size, SGD momentum) within the main text and appendix of the paper; see Section 6 and Appendix B. Our experiments use the publicly available CIFAR-FS dataset Bertinetto et al. (2018) and the standard ResNet-18 architecture He et al. (2016). Our experiment code is implemented using the JAX library and all experiments were conducted using Google Colab TPUs which are standardized and readily available hardware platforms.

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

## A    THEORETICAL GUARANTEES ON FINE-TUNING PREDICTIVE PERFORMANCE

Here we discuss theoretical guarantees on downstream predictive performance when employing the version of the downstream free energy strategy in equation 9. We would like to give an analysis of downstream predictive performance without being tied to a specific training algorithm e.g., SGD

with momentum, ADAM, etc. Towards this end, we consider measuring predictive performance through quantities related to the **downstream posterior distribution over neural network weights**:

$$p^1(w; w^*, \gamma) \propto \exp\{-m\mathrm{K}^1(w)\}\varphi_\gamma(w - w^*) \tag{15}$$

This does not mean we are advocating for Bayesian prediction, but rather we believe the posterior distribution above contains highly relevant information that all sensible downstream training algorithms are sensitive to.

Since fine-tuning entails finding a small perturbation of said $w^*$ which performs well on the downstream training dataset $\mathcal{D}^1$, we might consider an indicator of the downstream *training* performance to be given by

$$\mathrm{T}_m(w^*) := \mathbb{E}_{w \sim p^1(w; w^*, \gamma)} \hat{\mathrm{K}}^1(w). \tag{16}$$

Let us call equation 16 the **downstream Gibbs training error**. Let $w^*$ and $\gamma$ satisfy assumption 5.1. Then, on average, over the draw of $\mathcal{D}^1$, the expected downstream Gibbs training error is given by

$$\mathbb{E}_{\mathcal{D}^1} \mathrm{T}_m(w^*) = \mathrm{K}^1(w^{*1}) + \frac{\lambda^1(w^*) - \nu^1(w^*)}{m} + o\left(\frac{1}{m}\right) \tag{17}$$

where $\nu^1(w^*)$, like the local learning coefficient $\lambda^1(w^*)$, is a positive number called the *singular fluctuation* that is an invariant of the underlying model-truth-prior triplet. Since $\nu^1(w^*)$ is always positive, the strategy in equation 9 leads us to select a checkpoint that minimizes an upper bound on $\mathbb{E}_{\mathcal{D}^1} \mathrm{T}_m(w^*)$.

We can also look at the population counterpart to equation 16 given by

$$\mathrm{G}_m(w^*) := \mathbb{E}_{w \sim p^1(w; w^*, \gamma)} \mathrm{K}^1(w) \tag{18}$$

Let us call equation 18 the **downstream Gibbs test error**. The expected value of this, over the draw of $\mathcal{D}^1$ is given by

$$\mathbb{E}_{\mathcal{D}^1} \mathrm{G}_m(w^*) := \mathrm{K}^1(w^{*1}) + \frac{\lambda^1(w^*) + \nu^1(w^*)}{m} + o\left(\frac{1}{m}\right). \tag{19}$$

It does not appear the strategy in equation 9 gives control over the (expected) downstream Gibbs test error.

Finally consider the test error resulting from Bayesian model averaging:

$$\mathrm{G}_m^{\mathrm{BMA}}(w^*) := \mathbb{E}_{r^1(x)} D_{\mathrm{KL}}(r^1(y|x)||\mathbb{E}_{w \sim p^1(w; w^*, \gamma)} p(y|x, w)) \tag{20}$$

where the expectation over the posterior has been moved *inside* the logarithm. Let us call equation 20 the **downstream Bayes test error**. We have that

$$\mathbb{E}_{\mathcal{D}^1} \mathrm{G}_m^{\mathrm{BMA}}(w^*) := \mathrm{K}^1(w^{*1}) + \frac{\lambda^1(w^*)}{m} + o(\frac{1}{m}). \tag{21}$$

It is evident that the strategy in equation 9 leads us to select a checkpoint that minimizes an upper bound on $\mathbb{E}_{\mathcal{D}^1} \mathrm{G}_m^{\mathrm{BMA}}(w^*)$.

## B  EXPERIMENT DETAILS

This section provides details for the experiment results presented in Figure 1 and Figure 2. For these experiments we use the CIFAR-FS dataset (Bertinetto et al., 2018) which has been pre-partitioned into 64 meta-training classes, 14 meta-validation classes and 20 meta-test classes. Each class contains 600 examples. We use the meta-training dataset for pretraining and the meta-test dataset during fine-tuning. We do not use the meta-validation dataset.

**Random seeds**  To account for stochasticity, we repeat all experiments below with 5 different random seeds. These seeds control the randomness in the pretraining optimization trajectory, the train-test split and the fine-tuning optimization trajectory in full meta-test finetuning (Section D.2 below), and the construction of few-shot tasks in few-shot meta-test finetuning (Section D.2 below). The variability across the random seeds is reflected in Figure 2, although the error bands may not always be visible due to the wide scale of the $y$-axis in some cases.

## B.1 PRETRAINING DETAILS

We pretrain a ResNet-18 (He et al., 2016) on the CIFAR-FS meta-training dataset (Bertinetto et al., 2018) using SGD with cross-entropy loss. We vary SGD hyperparameters such as the learning rate, batch size, and momentum. We use plain SGD optimizer without any regularization nor schedule to avoid masking effects. We used random crop and random flip for data augmentation. Throughout training we report the pretraining train loss on the augmented data (Figure 2 first column) and the pretraining WBIC computed on the augmented data (Figure 2 second column). Note, we use the same SGLD hyperparameters to compute the WBIC across all experiments. That is, we use step size $\epsilon = 2 \times 10^{-7}$, chain length of 3,000 iterations, batch size of 2,048, $\gamma = 1.0$, and $\beta^* = \frac{1}{\log n}$ where $n$ is the size of the pretraining dataset.

**Learning rate.** For experiments that vary the learning rate in Figure 2 (top row), for each learning rate value in {0.01, 0.05, 0.1, 0.2} we run SGD without momentum with a fixed batch size of 512 for 50,000 iterations. The WBIC estimations were performed every 2,000 iterations with the SGLD hyperparameters above.

**Batch size.** For experiments that vary the batch size in Figure 2 (middle row), for each batch size in {16, 32, 64, 128, 256, 512} we run SGD without momentum with a fixed learning rate of 0.05 for 50,000 iterations. The WBIC estimations were performed every 4,000 iterations with the SGLD hyperparameters above.

**Momentum.** For experiments that vary the momentum in Figure 2 (bottom row), for each momentum in {0.0, 0.2, 0.4, 0.6, 0.8} we run SGD with a fixed learning rate of 0.01 and batch size of 512 for 80,000 iterations. The WBIC estimations were performed every 2,000 iterations with the SGLD hyperparameters above.

## B.2 FINE-TUNING DETAILS

We perform fine-tuning in two scenarios: full CIFAR-FS meta-test finetunining which uses all 20 classes of the meta-test set, and few-shot meta-test finetuning which consists of multiple tasks constructed from the CIFAR-FS meta-test dataset. In both settings we fine-tune a ResNet-18 model initializing the weights of the ResNet backbone with the pre-training weights. The weights of the model head are randomly initialized.

**Full meta-test fine-tuning.** When fine-tuning on the full CIFAR-FS meta-test dataset, we use all 20 meta-test classes and all 600 examples in each class. We then create an 80/20 train/test split. We use SGD with $L^2$ regularization rate of 0.01 and with a fixed learning rate of 0.0001 for the model backbone and a fixed learning rate of 0.01 for the model head. We fine-tune for 100 steps using a batch size of 128.

**Few-shot meta-test fine-tuning.** For few-shot fine-tuning, we use only part of the CIFAR-FS meta-test dataset by sampling 5-class classification tasks randomly from the 20 classes available in the meta-test dataset. For each of these 5 classes we sample 5 training examples to create a 5-shot dataset for fine-tuning. During fine-tuning, as with full meta-test fine-tuning, we use a fixed learning rate of 0.0001 for the model backbone and a fixed learning rate of 0.01 for the model head. We perform 100 steps of full-batch gradient descent (GD) with $L^2$ regularization rate of 0.001 and then measure the model performance on 100 random test samples from each class. This constitutes a single task. Finally, we report the resulting accuracy rates averaged over 100 randomly chosen tasks.

## C EXAMPLES OF PROPOSITION 5.3

In this section we provide two detailed examples involving Gaussian distributions which help to illustrate Proposition 5.3 in action.

**Example 1** (Covariate shift between pretraining and downstream distributions). *Suppose $r^0(y|x) = r^1(y|x) = r(y|x)$. Our pretraining and fine-tuning joint model is $p^i(x, y|w) = p(y|x, w)r^i(x)$.*

Then we have $\lambda^0(w^*) = \lambda^1(w^*)$ and $\mathrm{K}^i(w) = \mathbb{E}_{r^i(x)} K(x,w)$ where $K(x,w) = D_{\mathrm{KL}}(r(y|x)||p(y|x,w))$. Writing

$$\mathbb{E}_{r^1(x)} K(x,w) = \int K(x,w) \frac{r^1(x)}{r^0(x)} r^0(x)\, dx$$

we have that if $M = \max_{x \sim r^0(x)} \frac{r^1(x)}{r^0(x)} < \infty$ then

$$\mathbb{E}_{r^1(x)} K(x,w) \leq M \mathbb{E}_{r^0(x)} K(x,w)$$

Putting this together we have $D = 0$ and

$$\mathrm{K}^1(w^{*1}) \leq \mathrm{K}^1(w^*) \leq M\mathrm{K}^0(w^*).$$

Suppose the two covariate distributions are Gaussians

$$r^i(x) \propto \exp\{-\frac{||x - \mu_i||_2^2}{2\sigma_i^2}\}$$

then $M$ is finite if $\sigma_0 > \sigma_1$, in which case $M = \frac{\sigma_0}{\sigma_1} \exp\{\frac{(\mu_0 - \mu_1)^2}{2(\sigma_0^2 - \sigma_1^2)}\}$

**Example 2** (Nuisance parameter mismatch between pretrain and downstream distributions). *Suppose the pretrain ($i = 0$) and downstream ($i = 1$) distributions are given by*

$$r^i(x,y) = r(y|x, w_0, \sigma_i^2) r(x)$$

*where $r(y|x, w_0, \sigma_i^2) = N(f_{w_0}(x), \sigma_i^2)$ with $f_w(x)$ representing neural network with weight $w$. The pretraining and fine-tuning model are given by*

$$p^i(x,y|w) = r(y|x, w, \sigma_i^2) r(x)$$

*Then we have $\lambda^0(w^*) = \lambda^1(w^*)$ and $M = \sigma_0/\sigma_1$.*

# D ADDITIONAL EXPERIMENTS FOR MINI-IMAGENET; SEE FIGURE 3

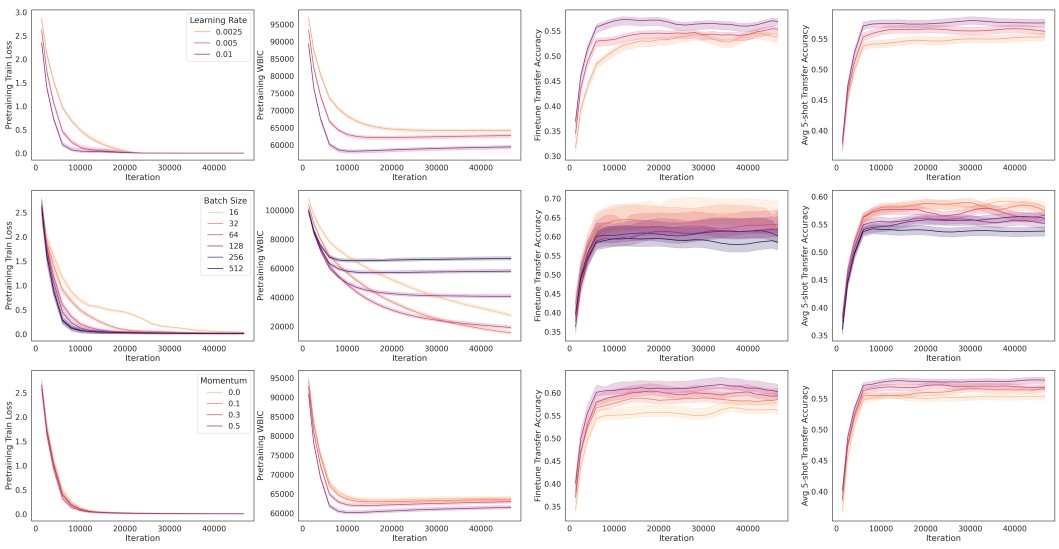

Figure 3: Model checkpoints with lower pretraining WBIC (second column) consistently result in better transfer accuracy, both when fine-tuning on the full downstream dataset (third column) and in the few-shot setting (fourth column). Lower pretraining WBIC correlates with better downstream performance for **Top row:** larger learning rates, **Middle row:** smaller batch sizes, and **Bottom row:** increased momentum.

### D.1 PRETRAINING DETAILS

We pretrain a VGG-16 (Simonyan, 2014) on the mini-Imagenet meta-training dataset (Dhillon et al., 2019) using SGD with cross-entropy loss. We vary SGD hyperparameters such as the learning rate, batch size, and momentum. We use plain SGD optimizer without any regularization nor schedule to avoid masking effects. We used random crop and random flip for data augmentation. Throughout training we report the pretraining train loss on the augmented data (Figure 2 first column) and the pretraining WBIC computed on the augmented data (Figure 2 second column). Note, we use the same SGLD hyperparameters to compute the WBIC across all experiments. That is, we use step size $\epsilon = 2 \times 10^{-7}$, chain length of 1,000 iterations, batch size of 1,024, $\gamma = 1.0$, and $\beta^* = \frac{1}{\log n}$ where $n$ is the size of the pretraining dataset. The results are plotted in Figure 3.

**Learning rate.**    For experiments that vary the learning rate in Figure 2 (top row), for each learning rate value in $\{0.0025, 0.005, 0.01\}$ we run SGD without momentum with a fixed batch size of 512 for 50,000 iterations. The WBIC estimations were performed every 2,000 iterations with the SGLD hyperparameters above.

**Batch size.**    For experiments that vary the batch size in Figure 2 (middle row), for each batch size in $\{16, 32, 64, 128, 256, 512\}$ we run SGD without momentum with a fixed learning rate of 0.01 for 50,000 iterations. The WBIC estimations were performed every 2,000 iterations with the SGLD hyperparameters above.

**Momentum.**    For experiments that vary the momentum in Figure 2 (bottom row), for each momentum in $\{0.0, 0.1, 0.3, 0.5\}$ we run SGD with a fixed learning rate of 0.005 and batch size of 512 for 50,000 iterations. The WBIC estimations were performed every 2,000 iterations with the SGLD hyperparameters above.

### D.2 FINE-TUNING DETAILS

We perform fine-tuning in two scenarios: full mini-Imagenet meta-test finetunining which uses all 20 classes of the meta-test set, and few-shot meta-test finetuning which consists of multiple tasks constructed from the mini-Imagenet meta-test dataset. In both settings we fine-tune a VGG-16 model initializing the weights of the VGG backbone with the pre-training weights. The weights of the model head are randomly initialized.

**Full meta-test fine-tuning.**    When fine-tuning on the full mini-Imagenet meta-test dataset, we use all 20 meta-test classes and all 600 examples in each class. We then create an 80/20 train/test split. We use SGD with $L^2$ regularization rate of 0.01 and with a fixed learning rate of 0.0001 for the model backbone and a fixed learning rate of 0.01 for the model head. We fine-tune for 500 steps using a batch size of 32.

**Few-shot meta-test fine-tuning.**    For few-shot fine-tuning, we use only part of the mini-Imagenet meta-test dataset by sampling 5-class classification tasks randomly from the 20 classes available in the meta-test dataset. For each of these 5 classes we sample 5 training examples to create a 5-shot dataset for fine-tuning. During fine-tuning, as with full meta-test fine-tuning, we use a fixed learning rate of 0.0001 for the model backbone and a fixed learning rate of 0.01 for the model head. We perform 100 steps of full-batch gradient descent (GD) with $L^2$ regularization rate of 0.01 and then measure the model performance on 100 random test samples from each class. This constitutes a single task. Finally, we report the resulting accuracy rates averaged over 100 randomly chosen tasks.

# E    COMPARISON OF WITH OTHER PRETRAINING METRICS

Recent work of Galanti et al. (2022) and Munn et al. (2024) examine the role of neural collapse and geometric complexity in transfer learning and suggest that this bias during pretraining towards low geometric complexity (and thus neural collapse) can help to explain the mechanisms behind the success of transfer learning. In short, their works shows that lower Geometric Complexity in a pre-trained embedding network promotes neural collapse on new target classes, simplifying the fine-tuning process and thus leading to improved downstream accuracy. As described in Section 2, one way to interpret their work is that the model geometric complexity or neural collapse can serve an an informative pretraining metric for assessing the quality of a model checkpoint with respect to success adaptation.

To assess the effectiveness of our free energy strategy in comparison to these other pretraining metrics, we conducted a correlation analysis, as summarized in Table 1. The Pearson correlation coefficients Pearson & Galton (1895) presented in the table were calculated using model checkpoints obtained from experiments with CIFAR-FS, trained on ResNet-18 to convergence.

These experiments, detailed in Section 6, involved a comprehensive exploration of the hyperparameter space. We swept across three hyperparameters (learning rate, batch size, and momentum), with six values for learning rate, six for batch size, and five for momentum. Each configuration was trained with five different random seeds, resulting in a total of 85 model checkpoints. For each checkpoint, we compared the Geometric Complexity, Neural Collapse, and Free Energy of the pretrained model to its downstream performance, measured via both full meta-test fine-tuning and few-shot meta-test fine-tuning. Notably, as indicated by the Pearson correlation coefficients in Table 1, the pretraining Free Energy exhibits a substantially stronger correlation with downstream performance than other metrics considered.

|  | Finetune Transfer Accuracy | Avg 5-shot Transfer Accuracy |
| --- | --- | --- |
| Geometric Complexity | $-0.767$ | $-0.443$ |
| Neural Collapse | $-0.632$ | $-0.1875$ |
| Free Energy | $\mathbf{-0.820}$ | $\mathbf{-0.8901}$ |

Table 1: Pearson correlation coefficients between pretraining metrics (geometric complexity, neural collapse and free energy) and downstream performance (finetune and few-shot transfer accuracy). The free energy has stronger correlation among all other pretraining metrics.

