# OpenReview forum: "Leveraging free energy in pretraining model selection for improved fine-tuning"
_ICLR.cc/2025/Conference — Submitted to ICLR 2025_

### Official Review · Reviewer_k33x · 2024-10-28

**Soundness:** 3
**Presentation:** 3
**Contribution:** 2
**Rating:** 5
**Confidence:** 4

**Summary:**

This paper investigates the adaptability of pretrained models through the lens of free energy. The authors validate the connection between downstream free energy and adaptability, subsequently proposing the concept of pretraining free energy, which relies solely on pretrained data. The effectiveness of this criterion in controlling downstream free energy is demonstrated, positioning it as a novel measure of downstream adaptability.

**Strengths:**

1. This paper presents a measure of downstream adaptability that relies solely on pretrained datasets.

2. The motivation of the method is clear and the manuscript is overall good.

**Weaknesses:**

1. The adaptability of pretrained models is often closely related to downstream tasks/datasets [1]. While this work proposes pretraining free energy as a general selection criterion, it lacks comparative analysis with prior research that typically utilizes a limited number of downstream dataset samples without access to pretrained data [1][2][3]. Such comparisons would strengthen the claims made in this paper.

2. There are concerns regarding the validity of the theoretical assumptions. Assumptions 5.1 and 5.2 do not specify their practical applicability or provide references for similar assumptions.

3. The experimental setup is limited, as validation experiments are conducted exclusively with ResNet-18 on the CIFAR-FS dataset. A broader exploration of various architectures and datasets would provide a more comprehensive evaluation of the proposed method.

[1] Not All Models Are Equal: Predicting Model Transferability in a Self-challenging Fisher Space

[2] Etran: Energy-based transferability estimation

[3] LEAD: Exploring Logit Space Evolution for Model Selection

**Questions:**

1. Intuitively, downstream adaptability is expected to vary across specific downstream task (e.g., between checkpoints A and B, adaptability may vary across tasks 1 and 2, where A performs better on task 1 and B on task 2, as referenced in past works). However, the proposed pretraining free energy seems to serve as a general  model selection criterion, raising questions about its rationale and necessity, since model selection is often focused on specific downstream tasks.

2. Assumption 5.2 appears overly generalized. It may not hold when two distributions overlap insufficiently or when higher-order statistical moments differ significantly. For example, in a simple case where $r_i(y|x)$ is the same,   $r_0(x)\sim N(0, 0.1), r_1(x)\sim N(0, 1)$, the described ratio diverges as x increases.

3. The distinction between the downstream free energy strategy proposed by the authors (line 245) and existing free energy criteria is unclear. If there is no substantive difference, Section 4.1 may be better suited as theoretical background rather than constituting a novel contribution of this paper.

4. The experimental results presented are insufficient. They rely on a single dataset and model architecture without exploring other transfer learning scenarios, such as cross-domain transfer learning, which would provide a more robust validation of the observation and proposed method.

---

> ### Author Response · Authors · 2024-11-21
>
> Thank you for your comments. We are happy that you found that the method clear and appreciate our measure of downstream adaptability which "relies solely on the pretraining datasets". We have addressed your comments and questions to the best of our ability below.
>
> Please consider raising your score or confidence if your concerns have been resolved. Thank you.
>
> **Weakness 1:**
> > "...it lacks comparative analysis with prior research..."
>
> Please see our top-level response on **Comparisons** at the top of this page.
>
> **Weakness 2:**
> > "There are concerns regarding the validity of the theoretical assumptions, Assumptions 5.1 and 5.2..."
>
> Assumption 5.1 is grounded in the original literature on Local Learning Coefficients (LLC), specifically regarding LLC estimation at local minima. Additionally, we noted on Line 330 that Assumption 5.2 follows from Yamazaki et al. (2007), which analyzed distribution shift scenarios. We believe these are rather mild assumptions actually.
>
> We have also included an additional statement in the manuscript addressing the feasibility of these Assumptions (see the subsection in blue titled "Interpretation and Feasibility of Assumption 5.2"). To provide more context, please also see our response to Reviewer SycF’s Weakness 4, where we further clarify Assumption 5.2 by explaining it in terms of distributional support—emphasizing the need for the pretraining distribution to cover a sufficiently large support relative to the downstream distribution.
>
> **Weakness  3 / Question 4:**
> > The experimental setup is limited
>
> Please see our top-level response on **Additional Experiments**. We have now included additional experiments to the manuscript that demonstrate the same relationship.
>
> **Question 1:**
> > ...the proposed pretraining free energy seems to serve as a general model selection criterion, raising questions about its rationale and necessity, since model selection is often focused on specific downstream tasks.
>
> Thank you for raising this question. We understand the concern about balancing a general model selection criterion with performance on specific downstream tasks. Our approach prioritizes generality, aiming to provide a selection criterion that applies across a wide range of potential downstream tasks. While tailoring checkpoints for specific tasks might improve performance in some cases, the pretraining free energy criterion is designed to work without such task-specific adjustments, making it practical and versatile in scenarios where downstream tasks may not be known during pretraining.
>
> **Question 2:**
> > Assumption 5.2 appears overly generalized. It may not hold when two distributions overlap insufficiently or when higher-order statistical moments differ significantly.
>
> In regards to the feasibility of Assumption 5.2 for real-world settings, we have included a subsection addressing the interpretation and feasiblity of this assumption (in blue in the updated manuscript).
>
> In short, we specifically focused on experimental settings where the pretraining dataset is much larger and more complex than the downstream dataset; cf. Kornblith et al., 2019. In our experiments, (and as reflected in practice) we achieved this by using pretraining datasets with a substantially larger set of image classes than the downstream dataset. We agree that if this were reversed; i.e., the pretraining dataset had substantially fewer classes than the downstream dataset, the relationship we establish in Prop 5.3 is uninformative. This would be similar to the example you provide where $r^0(x,y) \sim N(0, 0.1)$ and $r^1(x,y) \sim N(0, 1).$ Taken to this extreme, we also agree it would not make sense to apply our pretraining free energy selection criterion.
>
> **Question 3:**
> > The distinction between the downstream free energy strategy proposed by the authors (line 245) and existing free energy criteria is unclear.
>
> Note that we did not claim the free energy criterion for model selection is a novel contribution, and we highlight its wide usage in statistics in the "Relationship to Prior Work" where we discuss the distinction with the classic model selection criterion. Instead, the novelty and value of our work lies in applying this classic criterion to the area of pretraining and fine-tuning, where it has not been previously examined as a model selection tool for evaluating checkpoint adaptability. Our goal in Section 4.1 was to provide the necessary theoretical background for readers less familiar with free energy in the context of model selection.

---

> > ### Author Response · Authors · 2024-11-23
> >
> > As a quick followup…you stated in your Weakness 1 that the current work "lacks comparative analysis" and "such comparisons would strengthen the claims made in the paper". In addition to our response above, we have also conducted further analysis to quantitatively compare the pretraining free energy with the pretraining geometric complexity and neural collapse.
> >
> > To assess the relationship, we computed Pearson correlation coefficients between three pretraining metrics (geometric complexity, neural collapse, and free energy) and two downstream fine-tuning metrics (full fine-tuning transfer accuracy and average 5-shot transfer accuracy) utilizing our model checkpoints obtained from our CIFAR-FS experiments.
> >
> > As shown in the table below, pretraining Free Energy demonstrates a substantially stronger correlation with downstream performance compared to the other evaluated metrics.
> >
> > See Appendix E (in blue) in the updated version of the paper. Thank you!
> >
> > |                       | Finetune Transfer Accuracy | Avg 5-shot Transfer Accuracy |
> > |-----------------------|---------------------------|-----------------------------|
> > | Geometric Complexity | $-0.767$                    | $-0.443$                      |
> > | Neural Collapse      | $-0.632$                    | $-0.1875$                     |
> > | Free Energy          | $-\textbf{0.82}$                     | $-\textbf{0.8901}$                     |

---

> ### Comment · Reviewer_k33x · 2024-11-23
>
> Thank you for responding to my questions, they have resolved some of my concerns. However, the reply to Q2 cannot fully address my concerns.
>
> Theoretically, Assumption 5.2 remains challenging to satisfy in cases where the pretraining dataset contains more classes than the downstream dataset (Simple case: $r^0(x, y)\sim p_0(y=0)N(0,0.1)+p_0(y=1)N(1,0.1)+p_0(y=2)N(2,0.1), r^1(x,y)\sim p_1(y=0)N(0,1), p_0(y=i)=1/3, p_1(y=0)=1$). I recommend introducing additional constraints on this assumption, such as specifying certain statistical properties of the distributions, to enhance its general applicability. Otherwise, this assumption might not be broadly perceived as valid.
>
> Furthermore, I notice that Reviewer SycF raised W1, which also touches on the distributional differences between $r^0$ and $r^1$. I recommend including experimental results to validate this assumption. For example, you could use GMMs to estimate the feature distributions of the pretraining and downstream datasets as proxies for $r^i$, and verify whether the required distributional differences and proportions hold.

---

### Official Review · Reviewer_SycF · 2024-10-30

**Soundness:** 1
**Presentation:** 2
**Contribution:** 1
**Rating:** 3
**Confidence:** 4

**Summary:**

The authors propose to look at a free energy criterion to select the best possible pre-trained checkpoint for  later downstream tasks. They show a strong correlation between the proposed metric and the transfer accuracy on a downstream task. An theoretical derivation is made that claims that the pretraining free energy bounds the downstream free energy, which in turn gives a bound for the test error. A taylor expansion similar to the one in Lau et al. is used. Practically the free energy depends on the loss of the downstream dataset and a local learning coefficient, a way of measuring the complexity of a model. In a next step the downstream free energy is related to the pretraining free energy based on the work by Yamazaki et al., so that without access to the downstream data the best checkpoint can be determined for the downstream task.

**Strengths:**

* The paper aims to study a relevant problem, i.e. understanding when models will perform well on different data.
* I appreciate the quest for finding a theoretical basis, rather than simply performing millions of experiments to find a relation by accident.
* Overall, the paper is clear and in most parts easy to follow, even though it is mathematically a bit heavy, so that is not an easy task.

**Weaknesses:**

* I’m not confident that the relation that is given between the downstream energy and the pretraining free energy is very meaningful. It must rely on the similarity of both data distributions, as it is always possible to find a random distribution that has a much higher loss than that of the pretraining (e.g. change the labels). This relation is included in the comparison in proposition 5.3 by the quantity D. This relation is from Yamazaki et al, but importantly, they do not consider entirely different distributions. They look at how the test error is bounded by the training error when train and test distributions are different, but assume that both distributions have the same input domain. Especially when using two datasets that contain different classes, D will practically be infinite since: $r^0(y|x) = \frac{r^0(x, y)}{r^0(x)}$, and when $r^1(x, y) > 0 \implies r^1(x) > 0 \implies r^0(x) = 0$, when $r^0$ and $r^1$ are distributions over different domains. This is practically always true when considering image distributions. E.g. the domain of images of a car is completely different than that of images of a horse, unless they are both in the same image (which is not true for most image classification datasets). This not the case in Yamazaki et al., see for instance their numerical example in section 3.5, which looks at this quantity when $r^0$ and $r^1$ are normal distributions with a slightly different mean and standard deviation. $D$ is no longer used after line 353 because it is a constant, so it shouldn't be optimized. While that is true, constants shouldn't be ignored in the conclusion. E.g. when $f(x) < g(x) + 10e5$, optimizing $g$ instead of $f$ will bound $f$, but $f$ may still be as a large as $10e5$, even when $g(x) = 0$.


* There is no comparison to other measures that promise to do the same things. There are various other techniques in the related work (e.g. Liu et al, Galanti et al., Munn et al.), which should have been used to compare to. Similarly, there is no comparison to any other simple baseline, such as simply using the training loss.


* There are various observations made in Section 5.1 based on the proposed derivation. Although they seem plausible, there is no empirical validation of these observations. It would have been insightful to show examples where these observations are validated.


* On a high level, this relation says that if the pretraining error is low then the model will transfer better. This may be true when the pretraining dataset is larger and more complex than the downstream task, which other studies have also shown (e.g. [a]). This setting is also the one that is tested in the single experiment that is proposed, where the pretraining task is a larger part of CIFAR100. I do not believe that this relation is meaningful when the relation is reversed, and the pretraining task is significantly easier than the downstream tasks. There is only a single test of the proposed principle, with a single dataset configuration. Given the doubts I have with the theoretical validation (see above), the principle would need a lot more empirical proof to be convincing.


[a] Kornblith, S., Shlens, J., & Le, Q. V. (2019). Do better imagenet models transfer better?. In Proceedings of the IEEE/CVF conference on computer vision and pattern recognition (pp. 2661-2671).

**Questions:**

* Do you have an idea what the value of the constant $D$ is in practical scenarios, like the one you tested in Figure 2?
* Did you compare the proposed metric to other quantities the aim to predict how well a model transfers?
* Is there empirical validation of the observations in section 5.1?
* Did you test the proposed relation on more dataset (and combinations thereof) than the one currently in the paper?

---

> ### Author Response · Authors · 2024-11-15
>
> We would like to thank the reviewer for recognizing the relevance of our work, as well as for appreciating our theoretical approach. We are glad that the paper was generally clear and accessible, even with the mathematical depth required by our analysis.
>
> We are working diligently on point-by-point responses to all reviewers, but we wanted to immediately address your concerns about the general applicability of our approach. Below, we provide clarifications, which we hope will highlight the strength of our contributions and eventually encourage a reconsideration of the scores.
>
> **Theoretical concerns (Weakness 1 and 4)**
>
> We believe there may be a misunderstanding regarding the role of the constant  $D$, which we will first clarify. In Proposition 5.3, the constant $D$ is a very minor character. Recall our goal is to relate $mK^1(w^{\ast 1}) + \lambda^1(w^{\ast}) \log m$  to a quantity that only uses $K^0$ and $\lambda^0.$ We begin by writing $K^1(w) = f(w) + D,$ where $f(w)$ is the first expression in Line 344. We then establish $M K^0(w)$ as an upper bound on $f(w)$ which naturally extends to an overall upper bound on $K^1(w)$ that is $MK^0(w)+D.$
>
> Thus we establish that $K^1(w)=f(w)+D < MK^0(w) + D$. Our statement about disregarding the constant $D$ during optimization is in regards to the inequality $f(w) + D < MK^0(w) + D$. Note that the toy example you suggest where minimizing $g(x)=0$ but having $f(x) = 10e5$ simply cannot occur in our setting.
>
> Thank you for pointing out this confusion; we will clarify this point in the text.
>
> Next, we would like to address the real-world applicability of the relationship between pretraining and downstream free energy established in Proposition 5.3, as we believe this is your primary concern regarding the meaningfulness of our contribution.
> Your example of horse versus car images is a valuable thought experiment. You are correct that when $r^0(x,y)$ and $r^1(x,y)$ have disjoint label support, this would violate Assumption 5.2, which is specifically designed to prevent this situation by requiring controlled overlap between pretraining and downstream distributions. Specifically, if the support of the pretraining distribution $r^0$ is too small relative to the support of the downstream distribution $r^1$, the constant $M$ would become infinite, violating Assumption 5.2. Finally, you are correct that Yamazaki et al., particularly in their synthetic examples, actively avoid situations where Assumption 5.2 is violated.
>
> However, note that, as you correctly observe in your Weakness #4, in order to more reasonably satisfy Assumption 5.2, we specifically focused on experimental settings where the pretraining dataset is much larger and more complex than the downstream dataset. This setting has also been studied in prior work, as you noted (e.g., Kornblith et al., 2019). In our experiments, we achieved this by using pretraining datasets with a substantially larger set of image classes than the downstream dataset. We agree that if this is reversed; i.e., the pretraining dataset has substantially fewer classes than the downstream dataset, the relationship we establish in Prop 5.3 is uninformative. Taken to the extreme, we also agree it would be quite silly to apply our pretraining free energy selection criterion if the pretraining dataset contains only horse images and the downstream dataset contains only car images.
>
> In summary, we appreciate your insights, which have highlighted the importance of interpreting Assumption 5.2 with practical considerations in mind. We will incorporate this perspective into the paper to clarify its implications for real-world applications in the pretrain-then-adapt paradigm.

---

> > ### Author Response · Authors · 2024-11-21
> >
> > Thank you again for your comments. We are very happy that you found that the paper "clear and…easy to follow" and that you appreciate our approach towards this relevant problem of understanding the theoretical basis for transfer learning. Below we address your remaining comments to the best of our ability.
> >
> > Please consider raising your score or confidence if your concerns have been resolved. Thank you.
> >
> > **Weakness 2 / Question 2:**
> >  > "There is no comparison to other measures that promise to do the same things..."
> >
> > Please see our top-level response on **Comparisons.**
> >
> > **Weakness 3 / Question 3:**
> > > "...there is no empirical validation of these observations made in Section 5.1 based on the proposed derivation."
> >
> > The observations in Section 5.1 are theoretical conclusions derived directly from our formal theoretical analysis. As such, they are not empirical hypotheses but logical consequences of our theory, intended to provide interpretation and further clarify the theoretical implications.
> >
> > **Question 1:**
> > > "Do you have an idea what the value of the constant $D$...."
> >
> > The constant $D$, representing the KL divergence between the pretraining and downstream joint distributions, can be estimated with access to both pretraining and downstream data. However it is non-trivial to estimate $r^0(x,y)$ and $r^1(x,y)$ for the image datasets we use in our experiments.
> >
> > **Question 4:**
> > > "Did you test the proposed relation on more dataset..."
> >
> > We first note that your call for additional experimental settings seems to be motivated somewhat differently than the other reviewers. You wrote earlier in Weakness 4 that, “Given the doubts I have with the theoretical validation (see above), the principle would need a lot more empirical proof to be convincing.” We hope our clarifications above address these theoretical concerns, lessening the need, as you put it, for “performing millions of experiments to find a relation by accident.”
> >
> > Nonetheless, to address concerns raised by other reviewers regarding dataset and model variety, we completed a new experiment on mini-Imagenet for a VGG model which we have added in the Appendix which demonstrates the same relationship. Please see our top-level response on **Additional Experiments.**

---

> > > ### Author Response · Authors · 2024-11-23
> > > **Additional quantitative comparison with existing measures**
> > >
> > > You state in your Weakness 2
> > > > There is no comparison to other measures that promise to do the same things. There are various other techniques  which should have been used to compare
> > >
> > > In addition to our response above, we have also conducted further analysis to quantitatively compare the pretraining free energy with the pretraining geometric complexity and neural collapse.
> > >
> > > To assess the relationship, we computed Pearson correlation coefficients between three pretraining metrics (geometric complexity, neural collapse, and free energy) and two downstream fine-tuning metrics (full fine-tuning transfer accuracy and average 5-shot transfer accuracy) utilizing our model checkpoints obtained from our CIFAR-FS experiments.
> > >
> > > As shown in the table below, pretraining Free Energy demonstrates a substantially stronger correlation with downstream performance compared to the other evaluated metrics.
> > >
> > > See Appendix E (in blue) in the updated version of the paper. Thank you!
> > >
> > > |                       | Finetune Transfer Accuracy | Avg 5-shot Transfer Accuracy |
> > > |-----------------------|---------------------------|-----------------------------|
> > > | Geometric Complexity | $-0.767$                    | $-0.443$                      |
> > > | Neural Collapse      | $-0.632$                    | $-0.1875$                     |
> > > | Free Energy          | $-\textbf{0.82}$                     | $-\textbf{0.8901}$                     |

---

> > ### Comment · Reviewer_SycF · 2024-11-22
> >
> > Thank you for responding to my questions, they do solve some of the concerns I had. However, I'm still not convinced by the practicality of assumption 5.2 and $D$.
> >
> > It is true that I missed a step in my original comment, but I still believe that $D$ is important. The relation is $K^1(w) = f(w) + D < MK^0(w) + D$. Optimizing $K^0(w)$ bounds this relation, but $K^1$ can still be as large as $D$. Let's say that $K^0(w) = 0 \implies K^1(w) = f(w) + D < D \implies K^1(w) <  D $, hence still bounded by $D$.
> >
> > $D$ may still be arbitrarily lage, as you say yourselves: disjoint label support would violate assumption 5.2 and render a large $D$, and the baselines tested in the main paper still have disjoint label support even though the pretraining set is larger than the downstream task.
> >
> > I appreciate the comments on the comparison to other methods and the implicit comparison to the training loss. However, given that my most important weaknesses still stand, I don't think raising my scores would be adequate. The comparison between training loss and the proposed metric has convinced me that there may be value to the metric, but at this point it is very to grasp that idea (requiring multiple close reads for me at least). I believe the authors could significantly improve this paper by more clearly showing that simple metrics do not solve the problem and clearly highlight that the proposed metric improves it. I wouldn't rely on the theoretical justification as it is of right now, since assumption 5.2 is so loose it may practically be meaningless.
> >
> > I want to thank the authors for their effort again, as I definitely enjoyed thinking about this problem and the text.

---

> > > ### Author Response · Authors · 2024-11-23
> > >
> > > We appreciate the reviewer’s continued and thoughtful engagement with our work but believe there is a fundamental misunderstanding regarding the implications of the role of $D$ and its relationship with $K^1(w)$. Specifically:
> > >
> > > **The practicality and role of the constant $D$:**
> > >
> > > 1. **Misinterpretation of what happens when $K^0(w)=0$:**
> > >    With all due respect, the assertion that $K^1(w) \leq D$ when $K^0(w) = 0$ is incorrect. Note that, in this case, the derivation in fact implies $K^1(w) = D$. This result is stronger than the inequality suggested by the reviewer, indicating a slight misreading of the theoretical framework.
> > >
> > > 2. **Misplaced Concern About $K^1(w) = D$:**
> > >    The reviewer’s criticism that our theory allows $K^1(w)$ to be as large as $D$ overlooks the key interpretation of this term. $K^1(w)$ quantifies the KL divergence between the downstream data distribution and the model. A value of $K^1(w) = D$ does not invalidate our theory but rather reflects that the model $p(y|x, w)$ is not a perfect fit for $r^1(y|x)$. C’est la vie.
> > >
> > > 3. **Lack of Context in The Critique:**
> > >    The reviewer appears to imply that the bound $K^1(w) \leq MK^0(w) + D$ is meaningless if $K^1(w)$ can be as high as $D$. Again, it is important to emphasize that the potential for imperfect performance of the fine-tuned model on downstream data is an inherent aspect of transfer learning and, thus, par for the course. Our theoretical framework, however, offers a valuable tool for precisely quantifying the impact that this discrepancy between pretraining and downstream data distributions has on the utility of pretraining metrics for predicting downstream performance.
> > >
> > > We very much appreciate the reviewer's careful consideration of our work. However, we believe there may be a slight misunderstanding regarding the mathematical results and the interpretation and practical implications of $K^1(w)=D$ which has led to conclusions that are not substantiated by the theory we present.
> > >
> > >
> > > **Our experimental setup**
> > >
> > > The reviewer writes, “*the baselines tested in the main paper still have disjoint label support even though the pretraining set is larger than the downstream task*” which in turn violates Assumption 5.2 and thereby renders our theory meaningless.
> > >
> > > To clarify the scope of our theoretical findings, we note that our experimental setup is based on those used in the published works of Galanti et al. (ICLR) and Munn et al. (NeurIPS). While the scenario where pretraining and downstream datasets have disjoint label support may lie outside the strict assumptions of our theory (and, by extension, those in the cited works), we believe this does not mean the theory is useless. Instead, it highlights an important opportunity to explore the robustness of these theoretical predictions when assumptions are relaxed.
> > >
> > > Indeed, theoretical work often relies on simplifying assumptions to facilitate analysis. Experiments then serve as a crucial testing ground to assess the generalizability of theoretical insights when these assumptions are not fully met in practice. This is a common and valuable approach in both theoretical and empirical research.
> > >
> > > To draw a parallel, a linear regression model may yield meaningful results even when assumptions like homoscedasticity or Gaussian errors are not perfectly satisfied. The utility of a theory lies not solely in the literal fulfillment of its assumptions, but in the insights it generates and its predictive power in real-world scenarios.

---

### Official Review · Reviewer_iQYY · 2024-11-03

**Soundness:** 3
**Presentation:** 2
**Contribution:** 3
**Rating:** 6
**Confidence:** 4

**Summary:**

This paper proposes a novel free energy strategy for pretraining model selection to improve fine-tuning performance on downstream tasks. The work is grounded in extensive theoretical analysis, progressively examining the relationships between downstream task performance, downstream free energy, and pretraining free energy. It demonstrates that estimated pretraining free energy is a suitable proxy for selecting pretraining checkpoints without accessing downstream task data. Experiments are conducted under both full meta-test fine-tuning and few-shot meta-test fine-tuning settings, showing that strategies resulting in lower pretraining free energy (e.g., larger learning rates, smaller batch sizes, increased momentum) also yield better performance on downstream tasks.

**Strengths:**

1. The paper introduces a novel proxy, pretraining free energy, to identify the most suitable pretraining checkpoint for downstream tasks. This proxy does not rely on downstream task data, offering broader applicability.
2. The paper provides rigorous theoretical analysis, systematically proving constraints based on hypotheses, from downstream task performance to downstream free energy, and finally to pretraining free energy.
3. Based on the asymptotic pretraining free energy strategy, the paper provides some observations that would be helpful for pretraining.

**Weaknesses:**

1. The experimental section is overly simplistic, focusing solely on the CIFAR-FS dataset, without addressing cases where the downstream data distribution differs from the pretraining data distribution. Additionally, experiments use only a ResNet-18 model, which is relatively small in scale. Testing the theory on larger models, such as ViTs, would strengthen the study.
2. Previous work has explored pretrained model selection using proxies like neural collapse. The lack of comparison with existing methods weakens the persuasiveness of the results.
3. The paper’s presentation quality needs improvement; for instance, in Section 6, "Full meta-test fine-tuning" and "Few-shot meta-test fine-tuning" should be presented as parallel points. Additionally, consistency in cross-referencing formats and symbol representations is needed.

**Questions:**

1. The experiment section concludes that strategies such as larger learning rates, smaller batch sizes, and increased momentum yield better downstream transfer performance. However, these findings have already been established in prior work. Does the theory proposed offer additional insights or applications?
2. In L182, the paper assumes that the classification head $v$ of the pretraining task and $u$ of the downstream task share the same dimensionality. How is this assumption specifically applied in the theoretical analysis?
3. Can the proposed method be applied to unsupervised pretraining processes and situations where downstream tasks differ from the pretraining tasks?

---

> ### Author Response · Authors · 2024-11-21
>
> Thank you for your comments. We are very happy that you found the “rigorous theoretical analysis" insightful and find that our asymptotic pretraining free energy strategy could be "helpful for pretraining". As you point out, the pretraining free energy does "not rely on downstream task data" which we believe makes it a valuable and widely applicable proxy for assessing the quality of a pretraining checkpoint. We have addressed your comments to the best of our ability below.
>
> Please consider raising your score or confidence if your concerns have been resolved. Thank you.
>
> **Weakness 1:**
> > The experimental section is overly simplistic...
>
> Please see our top-level response on **Additional Experiments**. We have included additional experiments to the manuscript that demonstrate the same relationship.
>
> **Weakness 2:**
> > The lack of comparison with existing methods weakens the persuasiveness of the results.
>
> Please see our top-level response on **Comparisons.**
>
> **Weakness 3:**
> > "Full meta-test fine-tuning" and "Few-shot meta-test fine-tuning" should be presented as parallel points.
>
> Thank you. We have incorporated your suggestions on presenting “Full meta-test fine-tuning” and “Few-shot meta-test fine-tuning” in parallel and improving consistency in cross-referencing formats and symbol representations throughout the paper.
>
> **Question 1:**
> > Does the theory proposed offer additional insights or applications wrt learning rates, batch sizes, and momentum?
>
> While it’s true that prior work has suggested that larger learning rates, smaller batch sizes, and increased momentum can improve transfer performance, our contribution lies in verifying these findings within a rigorous theoretical framework. By using pretraining free energy as a selection criterion, we confirm that these strategies are indeed effective for improving downstream transferability.
>
> **Question 2:**
> > ...the paper assumes that the classification head $v$ of the pretraining task and $u$ of the downstream task share the same dimensionality. How is this assumption specifically applied in the theoretical analysis?
>
> Thank you for giving us the chance to clarify this. By assuming $u$ and $v$ are of the same dimensionality, we can unambiguously write $p(y|x,w)$ to refer to both the pretraining and fine-tuning model. This shows up in the rest of the theoretical development which only refers to $p(y|x,w)$.
>
> **Question 3:**
> > Can the proposed method be applied to unsupervised pretraining processes and situations where downstream tasks differ from the pretraining tasks?
>
> Yes absolutely, the proposed method can be applied to unsupervised pretraining. The pretraining free energy can be then defined for the model $p(x|w)$ rather than $p(y|x,w)$.

---

> > ### Author Response · Authors · 2024-11-23
> > **Additional quantitative comparison with existing measures.**
> >
> > As a quick followup…you stated in your Weakness 1 that
> > > the lack of comparison with existing methods like neural collapse weakens the persuasiveness of the results.
> >
> > To more directly address this, we have also conducted further analysis to quantitatively compare the pretraining free energy with the pretraining geometric complexity and neural collapse.
> >
> > To assess the relationship, we computed Pearson correlation coefficients between three pretraining metrics (geometric complexity, neural collapse, and free energy) and the two downstream fine-tuning metrics considered here (full fine-tuning transfer accuracy and average 5-shot transfer accuracy) utilizing our model checkpoints obtained from our CIFAR-FS experiments.
> >
> > As shown in the table below, pretraining Free Energy demonstrates a substantially stronger correlation with downstream performance compared to the other evaluated metrics.
> >
> > See also Appendix E (in blue) in the updated version of the paper. Thank you!
> >
> > |                       | Finetune Transfer Accuracy | Avg 5-shot Transfer Accuracy |
> > |-----------------------|---------------------------|-----------------------------|
> > | Geometric Complexity | $-0.767$                    | $-0.443$                      |
> > | Neural Collapse      | $-0.632$                    | $-0.1875$                     |
> > | Free Energy          | $-\textbf{0.82}$                     | $-\textbf{0.8901}$                     |

---

> > ### Comment · Reviewer_iQYY · 2024-11-25
> >
> > I would like to thank the authors for their efforts in addressing most of my concerns and questions, which has led me to raise my confidence to 4. However, I notice that the other two reviewers share a common concern regarding the relationship between the downstream energy and the pretraining free energy, which has also raised concerns on my end, particularly regarding the assumptions about data distribution in the theoretical analysis. Therefore, I have decided to maintain my current score.

---

### Author Response · Authors · 2024-11-21
**Additional Experiments and Comparisons with Other Measures.**

Thank you to the reviewers for your attention and thoughtful comments. Here we provide top-level response to address requests for additional experiments and comparison with other measures.

**Additional Experiments:**

We were able to complete a new experiment with VGG16 and mini-Imagenet. The results are in a new Appendix D of the manuscript. We see the same story unfold as our previous experiment with ResNet and CIFAR-FS: the pretraining free energy is highly correlated with fine-tuning performance.

**Comparisons:**

We thank iQYY and Sycf for their suggestion on comparing our method to other measures from Liu et al, Galanti et al, and Munn et al. We wish to first note that these studies do not compare their proposed measures with each other or with alternative methods. However, we have a more substantial objection to making a direct quantitative comparison with these methods. Specifically, Liu et al. (trace of Hessian), Galanti et al. (neural collapse), and Munn et al. (geometric complexity) all focus on a different fine-tuning approach: the linear probe. Our criterion is designed for settings that involve full fine-tuning, which represents a fundamentally different type of adaptation than the linear probe.

As to whether we can use the pretraining training loss as a simple baseline, we actually address this implicitly, as shown in the first column of Figure 2. We find that pretraining train loss often collapses to a similar value as training proceeds, rendering it ineffective for distinguishing different fine-tuning behaviors.

We appreciate K33x’s feedback (Weakness 1) on the need for comparative analysis with prior approaches that may utilize a limited number of downstream samples. However, our approach is intentionally designed to operate without any access to downstream data, setting it apart from methods that rely on such samples for selection criteria. This independence from downstream data access is precisely what enables the broader applicability of our criterion in scenarios where downstream information is either unavailable or unknown during pretraining.

As such, a direct comparison with methods requiring downstream samples would not align with our method's objective and could misrepresent its value, which lies in its adaptability assessment strictly from pre-training data. We believe this independence is a significant advantage in the context of model selection where downstream data might not always be accessible or practical to obtain.


**Summary of revisions in updated manuscript (highlighted in blue) in order of their appearance:**

- In our discussion of related works Liu et al, Galanti et al, and Munn et al, we now make it clear that these papers largely focus on fine-tuning performance associated to the linear probe
- Per Question 3 from iQYY we now briefly mention in the revision that the theory applies equally to the unsupervised setting
- Additional discussion on how to interpret Assumption 5.2 per the suggestions of SycF. See subsection "Interpretation and Feasibility of Assumption 5.2"
- We incorporated the suggestions by iQYY on presentation improvements (their Weakness 3)
- We now draw attention to the pretraining train loss as a simple baseline in the experimental section
- New Appendix D on new experiment VGG16 + mini-ImageNet

---

> ### Author Response · Authors · 2024-11-23
> **Additional correlation comparsion between free energy and other pretraining metrics on downstream performance.**
>
> As a followup to the note on **Comparisons** above, we have also conducted further analysis quantitatively comparing the pretraining free energy with the pretraining geometric complexity and neural collapse. Please see also a detailed section in the Appendix (see Appendix E, in blue) in the updated draft.
>
> In short, to assess the relationship between pretraining metrics and downstream performance, we computed Pearson correlation coefficients. These correlations were calculated between three pretraining metrics (geometric complexity, neural collapse, and free energy) and two downstream fine-tuning metrics (full fine-tuning transfer accuracy and average 5-shot transfer accuracy). We utilized model checkpoints obtained from our CIFAR-FS experiments, with models trained on ResNet-18 to convergence.
>
> As shown in the table below, pretraining Free Energy demonstrates a substantially stronger correlation with downstream performance compared to the other evaluated metrics.
>
>
>
> |                       | Finetune Transfer Accuracy | Avg 5-shot Transfer Accuracy |
> |-----------------------|---------------------------|-----------------------------|
> | Geometric Complexity | $-0.767$                    | $-0.443$                      |
> | Neural Collapse      | $-0.632$                    | $-0.1875$                     |
> | Free Energy          | $-\textbf{0.82}$                     | $-\textbf{0.8901}$                     |

---

> > ### Author Response · Authors · 2024-11-25
> >
> > We thank the reviewers for their thoughtful engagement with our work. We understand that perspectives differ on the balance between theoretical rigor and practical applicability in deep learning research. Our goal was to bridge this gap by introducing a method that performs well in realistic, experimentally relevant settings while offering theoretical insights grounded in assumptions that enable rigorous analysis.
> >
> > Our goal was to bridge this gap by introducing a theoretically motivated method that performs well for realistic, experimentally relevant settings and which offers additional insights grounded into the assumptions that enable rigorous analysis.
> >
> > We acknowledge that certain theoretical assumptions, such as Assumption 5.2, may not hold universally. These assumptions were necessary to provide rigorous guarantees in a field where developing theory for complex, real-world scenarios remains a significant challenge. While our theory does not apply if these assumptions are violated, our empirical results suggest that the pretraining free energy criterion nevertheless remains a robust and useful metric across various settings. We believe this reflects the broader value of our work in advancing understanding and informing future research.
> >
> > This discussion has also strengthened our paper and we sincerely thank the reviewers. Specifically, following reviewer feedback, we included comparisons with simpler baselines (Neural Collapse and Geometric Complexity), demonstrating stronger correlations between pretraining free energy and downstream performance. We also validated our approach on additional architectures and datasets (e.g., VGG + mini-ImageNet), further supporting its generality.
> >
> > We hope that the broader research community will find value in our contribution, both for its practical utility and its role in advancing theoretical discourse. We are grateful for this opportunity to engage in meaningful dialogue and will carry the insights from this process into future work.

---

### Meta-Review · Area_Chair_nFfc · 2024-12-15

**Metareview:**

While the paper presented an interesting concept with theoretical depth, it fell short in practical validation, experimental diversity, and comparative analysis. The key theoretical assumptions were deemed too generalized for real-world scenarios, limiting the paper's impact and applicability.

More specifically,

Assumption 5.2 regarding distributional similarity between pretraining and downstream tasks is overly generalized and not always realistic, particularly for disjoint label supports. This limitation undermines the broad applicability of the theory.

Experiments focus on the relatively small-scale model ResNet-18 and the tiny dataset CIFAR-FS. Testing on larger models and diverse datasets would strengthen the empirical validation. Cross-domain transfer scenarios are notably absent, limiting the generalizability of findings. While comparisons with neural collapse and geometric complexity were added, these are primarily statistical and lack deeper insights into practical implications.

**Additional Comments On Reviewer Discussion:**

Although the rebuttal addressed some concerns of Reviewer iQYY, iQYY found more concerns regarding the assumption relied on in their theoretical analysis after reading other reviews. Some clarification improved Reviewer SycF's appreciation of the paper, while SycF still believes the paper needs more justification and practical analysis to make it stand. And this conclusion also applies to Reviewer k33x.

---

### Decision · Program_Chairs · 2025-01-22

Reject